# The relationship between medical comorbidities and health-related quality of life among adults with type 2 diabetes: The experience of different hospitals in southern Bangladesh

**Adnan Mannan**[1]*, **Farhana Akter**[2], **Naim Uddin Hasan A. Chy**[3], **Nazmul Alam**[4], **Md. Mashud Rana**[5], **Nowshad Asgar Chowdhury**[6], **Md. Mahbub Hasan**[1]

1 Department of Genetic Engineering & Biotechnology, Faculty of Biological Sciences, University of Chittagong, Chattogram, Bangladesh, 2 Department of Endocrinology, Chittagong Medical College, Chattogram, Bangladesh, 3 Health Economics Research Group, Department of Economics, University of Chittagong, Chattogram, Bangladesh, 4 Department of Public Health, Asian University for Women, Chittagong, Bangladesh, 5 Department of Pharmacology and Therapeutics, Chittagong Medical College, Chattogram, Bangladesh, 6 Chattogram Diabetic General Hospital, Chattogram, Chittagong, Bangladesh

* adnan.mannan@cu.ac.bd

## Abstract

### Objective

Health-related quality of life (HRQoL) is a critical determinant to assess the severity of chronic diseases like diabetes mellitus. It has a close association with complications, comorbidities, and medical aid. This study aimed to estimate the prevalence of medical comorbidities and determine the relationship between comorbidities and HRQoL among type 2 diabetic patients of southern Bangladesh.

### Method

This study was a cross-sectional study conducted through face to face interviews using a pre-tested structured questionnaire and by reviewing patient's health records with prior written consent. The study was conducted on 2,136 patients with type 2 diabetes attending five hospitals of Chattogram, Bangladesh, during the tenure of November 2018 to July 2019. Quality of life was measured using the widely-used index of EQ-5D that considers 243 different health states and uses a scale in which 0 indicates a health state equivalent to death and 1 indicates perfect health status. The five dimensions of the quality index included mobility, self-care, usual activities, pain or discomfort, and anxiety or depression.

### Results

Patients with three comorbidities and with four or more comorbidities had a higher probability of reporting "extreme problem" or "some problem" in all five dimensions of the EQ-5D index compared with those without comorbidity (Odds ratio: mobility, 3.99 [2.72–5.87], 6.22

**Data Availability Statement:** All relevant data are within the paper and its Supporting information files.

**Funding:** This study was partially funded by "Special Allocation for Science and Technology", Ministry of Science & Technology, Government of the people's republic of Bangladesh (Award Number: 39.00.0000.009.14.019.21-745: 222 BS). The funders had no role in study design, data collection and analysis, decision to publish, or preparation of the manuscript. There was no additional external funding received for this study.

**Competing interests:** The authors have declared that no competing interests exist.

[3.80–10.19]; usual activity, 2.67 [1.76–4.06], 5.43 [3.28–8.98]; self-care, 2.60 [1.65–4.10], 3.95 [2.33–6.69]; pain or discomfort, 2.22 [1.48–3.33], 3.44 [1.83–6.45]; anxiety or depression, 1.75 [1.07–2.88], 2.45 [1.19–5.04]). The number of comorbidities had a negative impact on quality of life.

## Conclusion

Prevalent comorbidities were found to be the significant underlying cause of declined HRQoL. To raise diabetes awareness and for better disease management, the exposition of comorbidities in regards to HRQoL of people with diabetes should be considered for type 2 diabetes management schemas.

## 1. Introduction

Diabetes is a top-tier public health concern because of its ever-growing prevalence over the last few years. According to The International Diabetes Federation, approximately 463 million people globally currently have diabetes. As far as the projections go, by 2040, the count of people with diabetes will reach 700 million or double in a worst-case scenario [1]. Compared to the first world countries, developing countries have been found to have a higher prevalence rate of diabetes [2, 3]. This higher prevalence of diabetes in developing countries could be related to their population being less aware of the causes of diabetes as well as its chronic outcomes; which is why they are negligent of their lifestyle and compliance to health care advice resulting in the continuous rise of cases [3, 4].

In Bangladesh, a study published in the Bulletin of the World Health Organization revealed that 9.7% of the adult population (aged >35 years) have diabetes, and 22.4% are pre-diabetic [5]. During the tenure of 1995–2000, diabetes mellitus was found to be at 4% and during 2001–2005, the rate stood at 5% while it reached 9% by 2006–2010. For a developing country like Bangladesh, with limited resources in health care delivery, the increasing prevalence of diabetes is putting an impact on the economy as well [6]. The average annual expense of treatment and management of patients with diabetes in Bangladesh was estimated to be US$314, with the average direct cost being US$283 and the indirect being US$315 [7]. Apart from the economic standpoints, the mere existence of diabetes also takes a psychological toll on the individual diagnosed, due to this disease's chronic manifestations and auxiliary clinical complications [8–10]. The continuous restrictions to food and medications often render a patient's mind to think these efforts are futile, resulting in poor glycemic control, more complications and poor quality of life [11–13].

For diabetes and other chronic diseases, Health-Related Quality of Life (HRQoL) is a predominant proxy of lifestyle outcomes, medical treatment and quality of care. Findings obtained from assessing HRQoL can contribute to overall medical research as well as public health measures to minimize the spiking cases of chronic diseases [14, 15]. Previous studies demonstrated that poor HRQoL is associated with chronic illness and various comorbidities. For patients with diabetes, poor HRQoL can lead to fluctuating glycemic index, uncontrolled blood sugar level and trigger a chain of relevant and complicated comorbidities [16, 17]. A study conducted in the Netherlands found a high prevalence rate of comorbidities in patients with diabetes which was associated with the quality of life [18]. Another study from India also reported that the number of persisting comorbidities in patients with diabetes was inversely proportional to the HRQoL [19]. Most of these studies were conducted estimating the

prevalence of diabetes, associated risk factors and assessment of the quality of life. However, only limited studies highlighted the effect of comorbidities on quality of life among patients with diabetes [20].

To address and scrutinize HRQoL in patients with diabetes, several tools have been acclimatized. Among those, the EuroQol-5 Dimensions Questionnaire (EQ-5D) is a preference-based instrument due to its simplicity and reliability [21]. For evaluating clinical and economic outlooks of healthcare in population surveys, EQ-5D questionnaires are worthwhile because they provide a simple descriptive profile and a single index. Its utility was also reported by the Fenofibrate Intervention and Event Lowering in Diabetes study as an independent predictor of morbidity and mortality in patients with type 2 diabetes [21]. Different Asian countries, including Japan [22] and Korea [23], have already applied EQ-5D to evaluate HRQoL among their population with type 2 diabetes. Unlike other developing countries, in Bangladesh, the measurement of diabetes-related QoL using the EQ-5D instrument is sporadic and is carried out among limited samples using the 3L version [23–25].

To the best of our knowledge, this study pioneers in evaluating the relationship between HRQoL and comorbidities using the EQ-5D-5L instrument in the type 2 diabetes population in the southern regions of Bangladesh.

This study primarily aimed to estimate the prevalence of medical comorbidities and determine the relationship between comorbidities and HRQoL among type 2 diabetic patients of southern Bangladesh. We expect that the facts and findings of this study will facilitate developing further intervention on managing diabetes more rationally and coherently to further reduce diabetes incidences in Bangladesh and similar countries.

## 2. Methods

### 2.1 Study design, setting, and participants

This cross-sectional study was carried out among patients with type 2 diabetes attending five hospitals in Chittagong, Bangladesh between November 2018 to November 2019. All of these public and private hospitals and clinics (Chittagong Medical College Hospital, Chattogram Diabetic Hospital, Center for Specialized Care & Research (CSCR), Max hospital, and Chevron Diagnostics & Hospital) are located in South-Eastern part of Bangladesh and they provide treatment for around 2 million residents in Chittagong city and adjacent districts. A total of 2,136 patients with diabetes were selected consecutively from the outpatient departments of the selected hospitals. Using the formula n = (z^2 pq)/d^2, the minimum required sample size was determined; where z = 1.96, p = 31% (the expected proportion of problems in HRQoL), and d = 4% (permissible error of known prevalence) [23, 24]. The selection criteria that were set for recruiting study participants in this study were: adults men and women diagnosed with type 2 diabetes mellitus according to the WHO criteria, on oral medication for diabetes, registered at either of these selected hospitals or clinics hospital, referred by their attending physician, and a resident of Chittagong city or suburban areas. Patients with other types of diabetes and those who had serious illnesses requiring them to be hospitalized were excluded from the study.

### 2.2 Data collection and variables

We developed a semi-structured questionnaire for data collection. Before interviewing the study participants, the questionnaire was pilot tested. Face to face interviews were conducted by five study physicians, two research officers, and fifteen male and female research assistants. During data collection, random cross-checks were carried out by the principal investigator and co-principal investigators to ensure the quality of data. The questionnaire included

sections to collect data on socio-demographic characteristics, family history of diabetes, duration of diabetes, number of medications, self-reported comorbidities, and medication use. Anthropometric measurements of weight, height and body mass index (BMI) were measured using standardized protocols and calibrated equipment. Systolic and diastolic blood pressure (BP) was measured twice using digital BP monitors (Omron, SEM-1, Omron Corp., USA) at a 10-minute interval and the average of the two readings was used for this analysis. The time for the patient being diagnosed with diabetes, hypertension, and other co-morbidities were recorded from the self-reported questionnaires which were further confirmed by reviewing each participant's medical records that included general follow up prescriptions as well as biochemical assessment reports. Blood samples were collected using standard protocols and analyzed in the biochemistry laboratory for assessing glycated hemoglobin (HbA1c) levels. Keeping with the American Diabetes Association 2017 guidelines a patient was said to have controlled glycemic levels when the HbA1c was $\leq$ 7%and uncontrolled when the HbA1c was >7% [13, 26].

A structured, eight-item questionnaire (The EQ-5D-3L) was adapted to assess the HRQoL of the participants. The HRQoL records were self-reported and the questionnaire was already validated in different study settings. The measure of the HRQoL was done through a generic process to compare HRQoL in populations [27]. The EQ-5D-3L is a two-part questionnaire which includes: a health description system and Visual Analogue Scale (VAS). The health description system section records self-assessed health status according to five key dimensions: mobility, self-care, usual activities, pain/discomfort, and anxiety/depression. Each of the assessed dimensions is further split into three levels: no problem, some problems, and extreme problems. A total of 243 health states can be expressed by combining the different levels from each dimension.

### 2.3 Comorbidity measurement

Whether a subject has one or more comorbidity was assessed by self-reports made by the participants through yes/no responses to the questions implying "Has a doctor ever diagnosed that you had. . .". In the face-to-face interview, the trained interviewer verified the existence of chronic illnesses through checking diabetes notebooks of the patients, and laboratory diagnosis reports as well as the medications list of the participants to ensure the authenticity and comprehensiveness of the interview.

### 2.4 Data analysis

Categorical variables were analysed to estimate frequencies and proportions dividing the sample into two groups, with comorbidity and without comorbidity. The comorbid group included respondents suffering from at least one of the chronic conditions stated earlier, and those in the non-comorbid group were free from any chronic conditions other than diabetes. Chi-square test was used to compare categorical variables between groups. We then compared the proportions of patients with diabetes reporting having problems in the five dimensions of EQ-5D. In doing so, we merged the two levels of "some problems" and "extremes problems" together, which eventually gave us two categories of "having a problem" and "not having a problem" in the five dimensions. The percentages of patients experiencing the five-dimension-based problems and associated 95% confidence intervals were calculated. Next, the two-level problem dimensions were modelled using binary logit regressions to examine differences among the categories of different covariates including comorbidity. Last, given the continuous nature of the EQ-5D visual analogue scale, we used the Ordinary Least Square (OLS) technique to learn about the relationships between comorbidity and quality of life. As a robustness check,

we first estimated the model using only the main independent variable interest that is comorbidity. Then, we gradually added demographic variables and health-related variables to test the stability of the estimates. We used STATA/MP 14 (StataCorp LLC, Texas, USA) and GraphPadPrism(9.0, GraphPad Software, CA, USA) to perform the statistical analyses.

## 2.5 Ethical considerations

This study was approved by the Ethical Review Committee of Chittagong Medical College Hospital (CMC/PG/2019/57). Written informed consent, before the interview, was obtained from all participants. The objectives and procedures of the study were explained to the participants in their native language (Bengali).

## 3. Results

A total of 2,136 patients with type-2 diabetes participated in the study. Of them, information on all required variables was available for 1,978 respondents, which constituted the final sample of the study. Most participants (31.4%) were between 50–59 years of age (Table 1). Around 59% of respondents were female, and 42% studied up to the higher secondary level (12th class). The vast majority of the participants were married (90%), and more than half (52.3%) were homemakers. About 31.6% of participants came from a lower-middle-class family, earning between USD238 and USD417. Most respondents (41%) were overweight, with the BMI scores falling between 22.9 and 27. The distributions in all categorical variables except monthly family income and residence were different in the two groups of comorbid and non-comorbid patients. P-values, based on chi-square tests, were always found to be smaller than 0.05.

A significant share of the study patients (80.7%) suffered from at least one comorbid condition, with the average number of comorbidities being 1.6 (SD: 1.21). Fig 1 shows about 32% of them had a single comorbid condition, 27% had two comorbidities, 15% had three, and the remaining 7% suffered from four or more comorbidities. A higher percentage of female respondents reported more comorbidities than male respondents. 28.6% of female respondents reported two comorbidities as opposed to 25% of male. The percentages of female and male participants suffering from three comorbidities were 15.3% and 14.5%, respectively. The percentages of male and female patients with four or more comorbidities were 5.9% vs. 7.4%, respectively. Hypertension was found to be the most prevalent chronic condition (55%) among the patients with diabetes, followed by eye problem (53%), obesity (37%), and heart disease (20%) (Fig 1).

About 94% of respondents reported either "some problem" or "extreme problem" in at least one of the five dimensions of the EQ-5D measure namely mobility, self-care, usual activity, anxiety or depression, and pain or discomfort. When analyzed by comorbidities and gender, comorbid and female patients were found more likely to report "some problems" or "extreme problems" in the five dimensions (Fig 2). Table 2 below compares the proportions of patients reporting problems in the five dimensions. 85% of respondents reported "extreme problems" or "some problems" in the anxiety or depression dimension (CI: 83.4%-86.5%). The same table indicates the variables that are statistically significant based on estimated logit models. The results indicate a positive correlation between chronic conditions and the probability of experiencing problems in all five dimensions (Table 2). Respondents with three comorbidities and with four or more comorbidities had greater proportions of "extreme problems" or "some problems" in all five dimensions of the index compared with those without comorbidity (Odds ratio: mobility, 3.99 [2.72–5.87], 6.22 [3.80–10.19]; usual activity, 2.67 [1.76–4.06], 5.43 [3.28–8.98]; self-care, 2.60 [1.65–4.10], 3.95 [2.33–6.69]; pain or discomfort, 2.22 [1.48–3.33], 3.44 [1.83–6.45]; the anxiety of depression, 1.75 [1.07–2.88], 2.45 [1.19–5.04]). On the other

**Table 1. Characteristics of study patients suffering from diabetes.**

| | Total (n = 1,978) | | With comorbidity (n = 1,596) | | Without comorbidity (n = 382) | | p-value[1] |
|---|---|---|---|---|---|---|---|
| | N | % | N | % | N | % | |
| **Age (years)** | | | | | | | |
| 18–29 | 93 | 4.70 | 43 | 2.69 | 50 | 13.09 | ≤0.001 |
| 30–39 | 261 | 13.20 | 159 | 9.96 | 102 | 26.70 | |
| 40–49 | 511 | 25.83 | 411 | 25.75 | 100 | 26.18 | |
| 50–59 | 621 | 31.40 | 529 | 33.15 | 92 | 24.08 | |
| 60 and above | 492 | 24.87 | 454 | 28.45 | 38 | 9.95 | |
| **Gender** | | | | | | | |
| Female | 1169 | 59.10 | 970 | 60.78 | 199 | 52.09 | ≤0.001 |
| Male | 809 | 40.90 | 626 | 39.22 | 183 | 47.91 | |
| **Educational attainment** | | | | | | | |
| No education | 13 | 0.66 | 12 | 0.75 | 1 | 0.26 | 0.033 |
| Primary | 756 | 38.22 | 630 | 39.47 | 126 | 32.98 | |
| Up to Higher Secondary | 831 | 42.01 | 659 | 41.29 | 172 | 45.03 | |
| Bachelor and above | 378 | 19.11 | 295 | 18.48 | 83 | 21.73 | |
| **Marital status** | | | | | | | |
| Married | 1779 | 89.94 | 1431 | 89.66 | 348 | 91.10 | ≤0.001 |
| Never married | 40 | 2.02 | 24 | 1.50 | 16 | 4.19 | |
| Separated/divorced | 159 | 8.04 | 141 | 8.83 | 18 | 4.71 | |
| *Occupation* | | | | | | | |
| Business | 233 | 11.78 | 179 | 11.22 | 54 | 14.14 | ≤0.001 |
| Dependent | 63 | 3.19 | 59 | 3.70 | 4 | 1.05 | |
| Housewife | 1034 | 52.28 | 861 | 53.95 | 173 | 45.29 | |
| Others | 42 | 2.12 | 30 | 1.88 | 12 | 3.14 | |
| Retired | 191 | 9.66 | 177 | 11.09 | 14 | 3.66 | |
| Service | 388 | 19.62 | 269 | 16.85 | 119 | 31.15 | |
| Unemployed | 27 | 1.37 | 21 | 1.32 | 6 | 1.57 | |
| **Monthly family income** | | | | | | | |
| <USD238 | 549 | 27.76 | 434 | 27.19 | 115 | 30.10 | 0.382 |
| USD238-<USD417 | 625 | 31.60 | 501 | 31.39 | 124 | 32.46 | |
| USD417-<USD595 | 386 | 19.51 | 310 | 19.42 | 76 | 19.90 | |
| USD595-< USD893 | 233 | 11.78 | 191 | 11.97 | 42 | 10.99 | |
| USD893-< USD1,190 | 92 | 4.65 | 75 | 4.70 | 17 | 4.45 | |
| > = USD1,190 | 93 | 4.70 | 85 | 5.33 | 8 | 2.09 | |
| **Body Mass Index** | | | | | | | |
| Underweight | 33 | 1.67 | 22 | 1.38 | 11 | 2.88 | 0.013 |
| Normal | 399 | 20.17 | 307 | 19.24 | 92 | 24.08 | |
| Overweight | 810 | 40.95 | 655 | 41.04 | 155 | 40.58 | |
| Obese | 736 | 37.21 | 612 | 38.35 | 124 | 32.46 | |
| **Family history of diabetes** | 1086 | 54.90 | 851 | 53.32 | 235 | 61.52 | ≤0.005 |
| **Residence** | | | | | | | |
| Rural | 736 | 37.21 | 594 | 37.22 | 142 | 37.17 | 0.936 |
| Urban | 1242 | 62.79 | 1002 | 62.78 | 240 | 62.83 | |

[1] p-values were analyzed using chi-square tests, and p<0.05 was considered statistically significant.

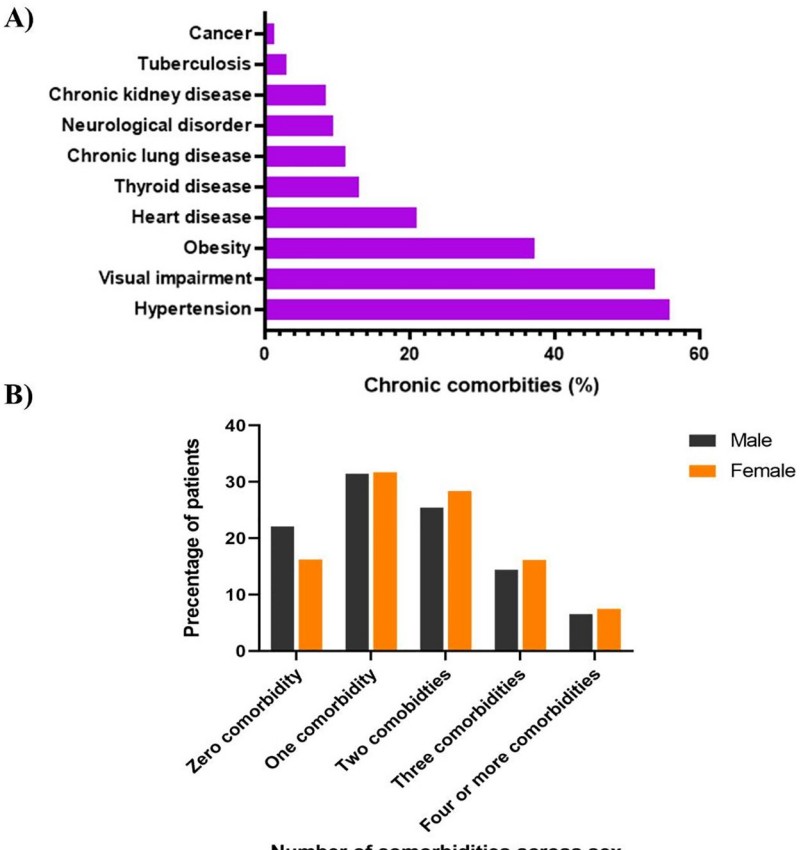

**Fig 1. Frequency of various comorbidities among diabetes patients of the southern part of Bangladesh.** A) Frequency of various comorbidities; B) Number of comorbidities present in males and females.

hand, participants suffering from two comorbid conditions showed a higher probability of having problems of mobility, pain or discomfort and of performing usual activity compared with those without any chronic conditions (Odds ratio: mobility, 2.35 [1.66–3.32]; usual activity, 1.87 [1.27–2.76]; pain or discomfort, 1.72 [1.24–2.38]). Respondents suffering from one comorbidity had a higher proportion of mobility problems compared with those without any chronic conditions (Odds ratio: mobility, 1.68 [1.20–2.35]). More than 30% of diabetes patients were found to have multimorbidities. Among the multimorbid patients, most common multimorbidities were hypertension and eye problem, hypertension and eye problem, hypertension and heart disease, hypertension and obesity, obesity and heart disease, hypertension and respiratory problem, Hypertension and kidney problem, eye and neurological problem together (S1 Table).

About 53% of both comorbid and non-comorbid patients experienced an average HRQoL, and around 23% of comorbid patients reported poor health quality as opposed to 9% of non-comorbid patients. The Pearson chi-square tests were performed to investigate the bivariate relationships between the medical comorbidities and a categorical measure of the quality of life with the three levels of poor, average, and good. The p-values for all tests except for the one between tuberculosis and quality of life were highly statistically significant ($\leq 0.001$). The results from the OLS regressions are presented in Table 3. The estimated coefficients on the various levels of comorbidities are significant at the 1% level. They are robust to changes in the

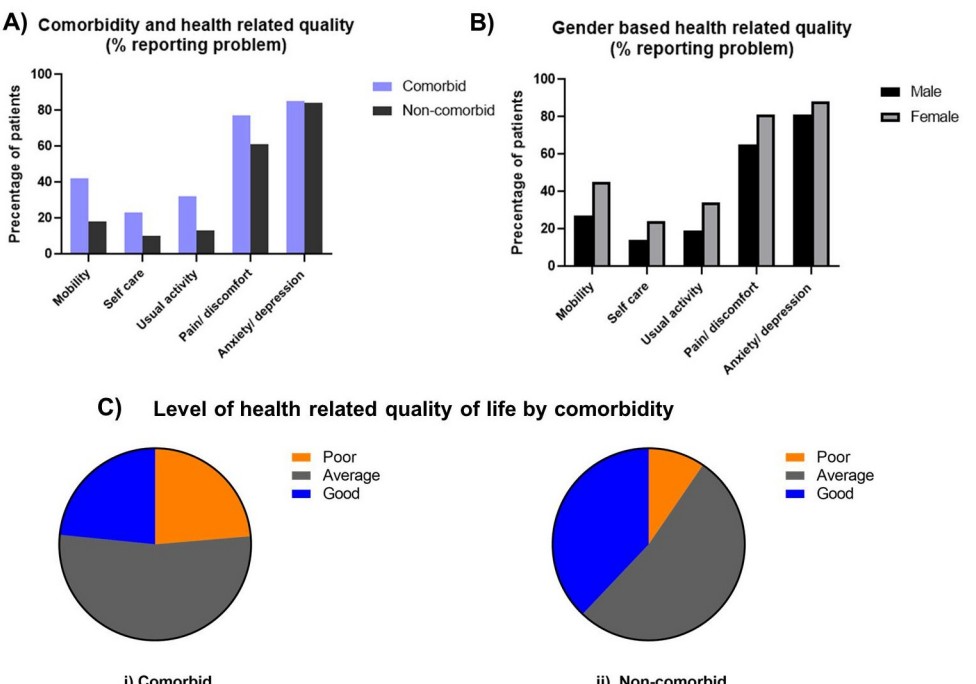

**Fig 2. Effect of comorbidity on health-related quality.** A) Comparison between the presence of various health-related problems among comorbid and non-comorbid T2D patients; B) Comparison between health related quality of life of male and female; C) HRQoL among comorbid and non-comorbid patients.

specifications even without compromising the statistical significance. The magnitudes of the coefficients increase with the levels of comorbidity while retaining strong estimated effects and hypothesized negative relationship with the quality of life index. Considering Model 1, Model 2, and Model 3 together, it can be seen that the least-square estimates on two comorbidities vary between -0.084 and -0.129, those on three comorbidities vary between -0.160 and -0.212, and those on four or more comorbidities vary between -0.269 and -0.329. While controlled for socio-demographic variables and various health-related characteristics, indicates that the quality of life is moderately diminished (-0.092) for the patients with two comorbidities than those without any comorbidity. HRQoL diminished for patients with three comorbidities and patients with four or more comorbidities by greater extents of -0.168 and -0.272, respectively than patients without comorbidities.

## 4. Discussion

Patients with diabetes mellitus often go through low levels of HRQoL because of the chronic nature of the disease and inadequate compliance followed over time. This study found out that more than two-thirds of the study participants had at least one comorbid condition, while HRQoL was found to be inversely correlated with having one or multiple comorbidities. This bilateral or multilateral correlation could be because people with low HRQoL may neglect their health and medical advice leading to further deterioration of their glycemic condition and other chronic medical problems including hyperlipidemia, obesity, etc [18, 27]. It is also possible that people with one or more comorbid conditions for example, having long term depression or hypertension may lead to developing other comorbidities, hence in turn deteriorates HRQoL.

**Table 2. Category-wise proportions of the five dimensions of EQ-5D measure.**

| | Mobility | Self-care | Usual activity | Pain/discomfort | Anxiety/depression |
|---|---|---|---|---|---|
| | % [95%CI] | % [95%CI] | %[95%CI] | % [95%CI] | % [95%CI] |
| **Total** | 36.4[34.3–38.6] | 19.4[17.7–21.2] | 26.7[24.8–28.7] | 73.5[71.5–75.4] | 85[83.4–86.5] |
| **Age (years)** | | | | | |
| 18–29 | 16.7 [10.2–26] | 6.7 [3–14.2] | 8.9 [4.4–17] | 48.9 [38.6–59.3] | 83.3 [74–89.8] |
| 30–39 | 21.8 [17.2–27.3] | 8.8 [5.9–12.9] | 12.3 [8.8–16.9] | 62.1 [56–67.8][S] | 82.4 [77.2–86.6] |
| 40–49 | 31.8 [27.9–35.9] | 13.9 [11.2–17.2] | 21 [17.7–24.7] | 72.7 [68.7–76.4][S] | 87.1 [83.8–89.7] |
| 50–59 | 36.6 [32.9–40.5] | 19.4 [6.4–22.7] | 27.7 [24.3–31.4][S] | 76.6 [73.1–79.8][S] | 86.5 [83.5–88.9] |
| 60 and above | 52.2 [47.8–56.6][S] | 33.1 [29.1–37.4][S] | 42.1 [37.8–46.5][S] | 81.1 [77.4–84.3][S] | 82.7 [79.1–85.8] |
| **Gender** | | | | | |
| Female | 43.3 [40.5–46.2] | 23.3 [20.9–25.8] | 32.6 [30–35.4] | 80 [77.6–82.2] | 87.7 [85.7–89.5] |
| Male | 26.4 [23.4–29.5][S] | 13.9 [11.6–16.4][S] | 18.1 [15.6–20.9] | 64.2 [60.9–67.5][S] | 81.1 [78.2–83.6][S] |
| **Educational attainment** | | | | | |
| No education | 46.2 [19.5–75.2] | 46.2 [19.5–75.2] | 46.2 [19.5–75.2] | 76.9 [42.8–93.7] | 84.6 [49–96.9] |
| Primary | 50.3 [46.7–53.8] | 27.5 [24.4–30.8] | 38.5 [35–42] | 78.6 [75.6–81.4] | 88.1 [85.5–90.2] |
| Up to Higher Secondary | 29.8 [26.8–33] | 16.2 [13.8–18.8] | 21.6 [18.9–24.5] | 74.3 [71.2–77.2] | 83.7 [81–86.1] |
| Bachelor and above | 22.8 [18.8–27.3] | 9.5 [7–13] | 13.5 [10.4–17.4] | 61.5 [56.5–66.3] | 81.7 [77.4–85.3] |
| **Marital status** | | | | | |
| Married | 34.3 [32.1–36.5] | 17.5 [15.8–19.3] | 25.1 [23.2–27.2] | 73 [70.9–75] | 84.7 [83–86.3] |
| Never married | 20.5 [10.3–36.8] | 10.3 [3.7–25.2] | 10.3 [3.7–25.2] | 56.4 [40–71.5] | 76.9 [60.5–87.9] |
| Separated/divorced | 64.2 [56.3–71.3][S] | 43.4 [35.8–51.3][S] | 47.8 [40.1–55.6] | 83.6 [77–88.7] | 89.9[84.1–93.8] |
| **Occupation** | | | | | |
| Business | 21.9 [17–27.7] | 10.7 [7.3–15.4] | 15.5 [11.3–20.7] | 64.8 [58.4–70.7] | 81.5 [76–86] |
| Dependent | 66.1[53.2–77] | 51.6 [39–64][S] | 58.1 [45.2–69.9][S] | 90.3 [79.7–95.7] | 95.2 [85.6–98.5] |
| Housewife | 43.7 [40.7–46.7] | 22.7 [20.3–25.4] | 33.3 [30.5–36.2] | 80.1 [77.5–82.4] | 87 [84.8–88.9] |
| Others | 28.6 [16.6–44.6] | 14.3 [6.3–29.1] | 14.3 [6.3–29.1] | 64.3 [48.2–77.7] | 71.4 [55.4–83.4][S] |
| Retired | 44 [37–51.2] | 27.2 [21.3–34] | 30.4 [24.2–37.3] | 77.5 [71–82.9] | 81.2 [74.9–86.1] |
| Service | 19.1 [15.5–23.3] | 8.2 [5.9–11.4] | 11.1 [8.3–14.6] | 58.2 [53.3–63.1] | 84.3 [80.3–87.6] |
| Unemployed | 22.2 [9.8–43] | 7.4 [1.7–27.2] | 14.8 [5.3–35.1] | 66.7 [46–82.5] | 74.1 [53.2–87.8] |
| **Monthly family income** | | | | | |
| <USD238 | 42.8 [38.7–47] | 24.5 [21.1–28.3] | 31.4 [27.7–35.5] | 74.4 [70.6–77.9] | 84.1 [80.8–86.9] |
| USD238-<USD417 | 35.4 [31.8–39.3][S] | 17.8 [15–21][S] | 25.2 [21.9–28.7][S] | 75.3 [71.8–78.6] | 85.9 [82.9–88.4] |
| USD417-<USD595 | 33.3 [28.8–38.2][S] | 18.2 [14.7–22.4][S] | 26.3 [22.1–31] | 72.4 [67.7–76.7] | 86.7[82.9–89.8] |
| USD595-< USD893 | 30.5 [24.9–36.7][S] | 15.9 [11.7–21.2][S] | 24.9 [19.7–30.9] | 70 [63.7–75.5] | 83.7 [78.3–87.9] |
| USD893-< USD1,190 | 34.8 [25.6–45.2] | 13 [7.5–21.8][S] | 20.7 [13.5–30.3][S] | 72.8 [62.7–81.1] | 84.8 [75.7–90.9] |
| > = USD1,190 | 34.4 [25.3–44.8] | 20.4 [13.3–30] | 20.4 [13.3–30] | 71 [60.8–79.4] | 80.6 [71.2–87.6] |
| **Body Mass Index** | | | | | |
| Underweight | 46.9 [29.8–64.8] | 21.9 [10.3–40.4] | 31.3 [17.1–50] | 84.4 [66.3–93.7] | 90.6 [73.3–97.1] |
| Normal | 31.4 [27–36.2][S] | 19.6 [16–23.8] | 26.9 [22.7–31.5] | 68.6 [63.8–73][S] | 86.4[82.7–89.5] |
| Overweight | 33.1 [30–36.5] | 17.4 [15–20.2] | 24 [21.2–27.1] | 73.2 [70–76.1] | 84.4 [81.8–86.8] |
| Obese | 42.2 [38.7–45.9] | 21.4 [18.6–24.5] | 29.3 [26.1–32.7] | 76.2 [72.9–79.1] | 84.6 [81.8–87] |
| **Family history of diabetes** | 34.2 [31.5–37.1] | 17.6 [15.5–20] | 24.7 [22.2–27.4] | 72.5 [69.8–75.1] | 85.3 [83.1–87.3] |
| **No. of comorbidities** | | | | | |
| 0 | 17.6 [14.1–21.8] | 10 [7.3–13.4] | 12.9 [9.8–16.6] | 60.6 [55.6–65.4] | 84 [79.9–87.4] |
| 1 | 29.5 [26.1–33.2][S] | 13.2 [10.8–16.1] | 20.4 [17.4–23.8] | 69.2 [65.5–72.7] | 80.5 [77.2–83.5] |
| 2 | 40.4 [36.3–44.6][S] | 20 [16.8–23.6] | 29.2 [25.5–33.2][S] | 78.7 [75–82][S] | 86.9 [83.8–89.5] |
| 3 | 53.2 [47.5–58.9][S] | 31.5 [26.4–37.1][S] | 38 [32.6–43.7][S] | 82.7 [77.9–86.6][S] | 89.2 [85–92.2][S] |
| > = 4 | 68.9 [60.5–76.2][S] | 45.9 [37.6–54.5][S] | 60 [51.4–68][S] | 89.6 [83.2–93.8][S] | 91.9 [85.8–95.5][S] |

(*Continued*)

**Table 2.** (Continued)

|  | Mobility | Self-care | Usual activity | Pain/discomfort | Anxiety/depression |
|---|---|---|---|---|---|
|  | % [95%CI] | % [95%CI] | %[95%CI] | % [95%CI] | % [95%CI] |
| **Residence** |  |  |  |  |  |
| Rural | 35.8 [33.1–38.5] | 18.3 [16.3–20.6] | 25.3 [23–27.8] | 73 [70.5–75.4] | 84.4 [82.3–86.3] |
| Urban | 37.5 [34–41] | 21.3 [18.4–24.4][S] | 28.9 [25.7–32.3] | 74.4 [71.1–77.4] | 86 [83.3–88.3] |

CI: Confidence interval, S = Significant at the 5% level.

The first category of each covariate was considered as the reference category in the logistic regressions.

A higher percentage of female respondents reported comorbidities than the male respondents. This finding is consistent with many studies conducted in Bangladesh and elsewhere [27–29]. Hypertension was found to be the most prevalent chronic condition among patients with type-2 diabetes, followed by eye complications, obesity, and heart disease. A systematic review study [30] reported higher rates of hypertension than this study among people with diabetes from Sweden [31], Germany [32], UK [33], Israel [34], Saudi Arabia [35], and Brazil [36], but lower in India [37], Taiwan [38], Mexico [39], and Japan [40]. More than three fourth of the patient with diabetes in this study were either overweight or obese while more than half of them had a family history of diabetes.

One or more HRQoL concerns related to mobility, self-care, routine activity, anxiety/depression, or pain/discomfort were found to be universally common among the study participants. Anxiety or depression was found to be the most common quality of care issue across all the socio-demographic categories by age, sex, educational attainment, marital status, and occupation. Noteworthy to mention the patients with no reported comorbidity had an almost similarly high burden of anxiety/depression across all the socio-demographic categories. A high level of depression among patients with diabetes was found to be associated with chronic stress that stemmed from poor glycemic control, fear of developing chronic complications, sleep deprivation, lack of physical exercise and diet [41–45]. The second most common quality of care concern was pain or discomfort while the least reported quality of care issue was related to self-care. Self-care is a very important quality attribute for patients with diabetes, there are several key elements of it relating to diet, physical activity, glycemic control, and medication [13, 46]. Self-care was reported as a less concerning issue by patients of younger age, male sex, unmarried individuals, and those without comorbidities. A similar finding of a high level of anxiety/depression and less self-care related problems was also reported by another recent study conducted in Bangladesh [28].

All five HRQoL issues were found to be commonly prevalent among patients with multiple comorbid conditions compared to the patients who reported no comorbid condition as found in a robust linear regression model. An inverse relationship was observed between the presence of comorbidities and HRQoL index, which showed a strong trend as estimated by least-square coefficients. After controlling for socio-demographic variables including age, sex, education, income threshold, place of residence, and biomedical factors including BMI and systolic and diastolic blood pressure, it was found that the quality of life moderately diminished for the patients with two comorbidities than those without any comorbidity; while the quality of life even progressively diminished for patients with four or more comorbidities compared to the patients without comorbidities. This finding has been supportive of other studies conducted in western countries [47–49]. Another study reported a high prevalence of comorbidities among diabetes patients in Bangladesh but they did find a significant relationship with HRQoL [25].

This study had a few limitations. Firstly, data collected through cross-sectional design are not meant for predicting any causal relationship between comorbidities and HRQoL. Study

**Table 3. Predictors of health-related quality of life.**

| | (1) | (2) | (3) |
|---|---|---|---|
| | **Model l** | **Model 2** | **Model 3** |
| **Comorbidity** | | | |
| No comorbidity (reference) | | | |
| One comorbidity | -.059*** | -.027 | -.033 |
| | (.02) | (.02) | (.02) |
| Two comorbidities | -.129*** | -.084*** | -.092*** |
| | (.021) | (.021) | (.022) |
| Three comorbidities | -.212*** | -.16*** | -.168*** |
| | (.023) | (.024) | (.025) |
| Four or more comorbidities | -.329*** | -.259*** | -.272*** |
| | (.03) | (.031) | (.032) |
| **Age (years)** | | | |
| 18–29 (reference) | | | |
| 30–39 | | -.089** | -.095** |
| | | (.039) | (.04) |
| 40–49 | | -.095** | -.103*** |
| | | (.038) | (.039) |
| 50–59 | | -.119*** | -.122*** |
| | | (.038) | (.039) |
| 60 and above | | -.158*** | -.156*** |
| | | (.041) | (.041) |
| **Gender** | | | |
| Female (reference) | | | |
| Male | | .121*** | .112*** |
| | | (.028) | (.028) |
| **Educational attainment** | | | |
| No education (reference) | | | |
| Primary | | -.057 | -.068 |
| | | (.075) | (.089) |
| Higher Secondary | | -.02 | -.035 |
| | | (.076) | (.09) |
| Bachelor | | .018 | .002 |
| | | (.078) | (.092) |
| **Marital status** | | | |
| Married (reference) | | | |
| Never married | | -.037 | .007 |
| | | (.055) | (.057) |
| Widowed/separated | | -.066** | -.079*** |
| | | (.026) | (.027) |
| **Occupation** | | | |
| Business (reference) | | | |
| Dependent | | -.08* | -.075 |
| | | (.048) | (.049) |
| Housewife | | .02 | .012 |
| | | (.035) | (.036) |
| Others | | -.017 | -.024 |
| | | (.05) | (.051) |

(*Continued*)

**Table 3.** (Continued)

| | (1) | (2) | (3) |
|---|---|---|---|
| | **Model l** | **Model 2** | **Model 3** |
| Retired | | -.016 | -.022 |
| | | (.032) | (.032) |
| Service | | .01 | .009 |
| | | (.026) | (.026) |
| Unemployed | | -.067 | -.033 |
| | | (.058) | (.065) |
| **Family income** | | | |
| <USD238 (reference) | | | |
| USD238-<USD417 | | .026 | .026 |
| | | (.018) | (.018) |
| USD417-<USD595 | | .022 | .018 |
| | | (.021) | (.021) |
| USD595-< USD893 | | .051** | .047* |
| | | (.025) | (.025) |
| USD893-< USD1,190 | | .005 | -.012 |
| | | (.035) | (.036) |
| > = USD1,190 | | .014 | .006 |
| | | (.036) | (.036) |
| **Residence** | | | |
| Rural (reference) | | | |
| Urban | | .047*** | .041*** |
| | | (.014) | (.014) |
| **Family history of diabetes** | | | |
| No (reference) | | | |
| Yes | | | .007 |
| | | | (.014) |
| **Body Mass Index** | | | |
| Underweight (Reference) | | | |
| Normal | | | .166*** |
| | | | (.057) |
| Overweight | | | .184*** |
| | | | (.057) |
| Obese | | | .163*** |
| | | | (.057) |
| Systolic blood pressure | | | .001 |
| | | | (.001) |
| Diastolic blood pressure | | | .001 |
| | | | (.001) |
| Constant | .677*** | .684*** | .352*** |
| | (.016) | (.09) | (.135) |
| Observations | 2136 | 2042 | 1978 |
| R-squared | .077 | .147 | .154 |

Standard errors are in parentheses.

*** p<0.01,

** p<0.05,

* p<0.10.

samples recruited from health care facilities may limit the generalizability of the study findings to diabetes patients. However, it is recommended for patients with diabetes to visit a health centre for check-ups at a regular interval. Self-reported data on several comorbidities and HRQoL indicators may introduce misclassification bias to a certain extent but the chances were minimal because the data collectors included medical graduates and reliance on diabetes record books were consulted when data were available. However, this study was unique since this is the first study conducted among patients with diabetes to study the effect of comorbidities on HRQoL and that the study samples were recruited from five health centres of government and private nature to increase the high variability of the study samples. The study used robust models fitted at 3 levels with gradual inclusion of only comorbidities, and sociodemographic variables and finally all-inclusive model comorbidities, sociodemographic variables, and key biomedical indicators.

This study underscores two-thirds of the study participants with diabetes mellitus suffer from one or more comorbidities. Their HRQoL inversely correlated with having multiple comorbidities along with diabetes. Hypertension was found to be the most prevalent chronic condition among patients with diabetes; a significant risk factor of cardiovascular diseases leading to stroke and disability. The findings of this study are consistent with other studies nationally and internationally. It is thus paramount to raise awareness among patients with diabetes to maintain a healthy lifestyle and follow health care recommendations to avoid or delay the development of other health complications to maintain good HRQoL. It is equally important for people with diabetes to control blood sugar and cholesterol level in their daily life through taking high fiber diet and food with low glycemic index.

## 5. Conclusion

This study provided robust estimates on the prevalence of chronic and acute multimorbidities, and the relationships between different multimorbidity patterns with HRQoL among people with type 2 diabetes in southern Bangladesh. These findings have significant implications for identifying patients with diabetes at higher risk of experiencing a lower quality of life and demonstrated the importance of early identification and treatment of diseases. The study also provides useful evidence for decision-makers upon optimizing the allocation of health resources, and intervention strategies for the health administrative departments to strengthen the management and monitoring of chronic diseases, and raise awareness of the prevention of chronic diseases. Future research should explore the causal relationship of multimorbidities with the quality of life in a prospective cohort study.

## Supporting information

**S1 Table. Most common multi-morbidities among diabetes patients of southern Bangladesh.**
(DOCX)

**S1 Questionnaire. Understating the health related quality of life and molecular the patients in the southern part of Bangladesh.**
(PDF)

## Acknowledgments

Authors would like to thank the research assistants of Disease Biology and Molecular Epidemiology Research Group, Chattogram for their support. Special thanks to the authorities of Chattogram Medical College hospital, Centre for Specialized Care & Research (CSCR) Hospital,

Max Hospital, Chevron Hospitals and Chattogram Diabetic General Hospital for their unconditional support.

## Author Contributions

**Conceptualization:** Adnan Mannan, Farhana Akter, Md. Mashud Rana, Nowshad Asgar Chowdhury, Md. Mahbub Hasan.

**Data curation:** Adnan Mannan, Farhana Akter, Naim Uddin Hasan A. Chy, Md. Mashud Rana, Md. Mahbub Hasan.

**Formal analysis:** Naim Uddin Hasan A. Chy, Md. Mahbub Hasan.

**Investigation:** Adnan Mannan, Farhana Akter, Md. Mashud Rana.

**Methodology:** Nazmul Alam.

**Project administration:** Adnan Mannan, Md. Mashud Rana, Nowshad Asgar Chowdhury.

**Resources:** Nowshad Asgar Chowdhury.

**Software:** Naim Uddin Hasan A. Chy.

**Supervision:** Adnan Mannan.

**Validation:** Naim Uddin Hasan A. Chy, Nazmul Alam.

**Visualization:** Nazmul Alam.

**Writing – original draft:** Adnan Mannan, Nazmul Alam.

**Writing – review & editing:** Nazmul Alam.

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
