## [Decision Letter · Decision Letter 0]

19 Jan 2022

PONE-D-21-37788Effect of comorbidities on health-related quality of life among adults with type 2 diabetes attending different hospitals in southern BangladeshPLOS ONE

Dear Dr. Mannan,

Thank you for submitting your manuscript to PLOS ONE. After careful consideration, we feel that it has merit but does not fully meet PLOS ONE’s publication criteria as it currently stands. Therefore, we invite you to submit a revised version of the manuscript that addresses the points raised during the review process.

We look forward to receiving your revised manuscript.

Kind regards,

Mohammad Farris Iman Leong Bin Abdullah, Dr Psych

Academic Editor

PLOS ONE

Journal Requirements:

(No)

Additional Editor Comments:

1. Please ensure that your manuscript meets PLOS ONE's style requirements, including those for file naming and tables. The PLOS ONE style templates can be found at:

2. Please ensure that your tables are formatted according to PLOS ONE's style requirements which can be found at:

https://journals.plos.org/plosone/s/tables

3. Please seek assistance from a native English speaker or professional English editing service to proofread your revised manuscript as there are some grammatical errors which need to be corrected.

Reviewers' comments:

Reviewer's Responses to Questions

**Comments to the Author**

1. Is the manuscript technically sound, and do the data support the conclusions?

Reviewer #1: No

Reviewer #2: Yes

2. Has the statistical analysis been performed appropriately and rigorously? 

Reviewer #1: Yes

Reviewer #2: I Don't Know

3. Have the authors made all data underlying the findings in their manuscript fully available?

Reviewer #1: Yes

Reviewer #2: Yes

4. Is the manuscript presented in an intelligible fashion and written in standard English?

Reviewer #1: Yes

Reviewer #2: Yes

5. Review Comments to the Author

Reviewer #1: Effect of comorbidities on health-related quality of life among adults with type 2 diabetes attending different hospitals in southern Bangladesh

Title: The title is misleading because there is an implication of the language of causality. Please rephrase the title to: “The relationship between medical comorbidities and health-related quality of life among adults with type 2 diabetes: The experience of different hospitals in southern Bangladesh”.

Overall comments:

The writing of the manuscript is good, with fair scientific merits based on the statistical methods employed. Having said that, there were some concerns of inference on the statistics of “correlation” when the author(s) mentioned that “The number of comorbidities was found to be negatively correlated with the HRQoL. “ Has any correlation study been employed in this research? The authors may change the wording of the sentences or employ the relevant statistics and discuss the findings.

Regarding the use of the statistical inference, please spell out OR as an odds ratio. The 95% CI was missing at each OR. Please insert it and standardise it through the text.

Specific comments:

Regarding abstract:

The abstract needs some revisions, as it contains the language of causality in the objective. The writing style was also required to be refined. I suggest some changes:

"Objective Health-related quality of life (HRQoL) is a critical determinant to assess the severity of chronic diseases like diabetes mellitus. It has a close association with complications, comorbidities, and medical aid. This study aimed to estimate the prevalence of medical comorbidities and determine the relationship between comorbidities and HRQoL among type 2 diabetic patients of southern Bangladesh. Method: This study was a cross-sectional study conducted through face-to-face interviews using a pre-tested structured questionnaire and by reviewing patient’s health records with prior written consent. The study was conducted on 2,136 patients with type 2 diabetes attending five hospitals of Chattogram, Bangladesh, during the tenure of November 2018 to July 2019. Quality of life was measured using the widely-used index of EQ-5D that considers 243 different health states and uses a scale in which 0 indicates a health state equivalent to death and 1 indicates perfect health status. The five dimensions of the quality index included mobility, self-care, usual activities, pain or discomfort, and anxiety or depression. Results: Patients with three comorbidities and with four or more comorbidities had a higher probability of reporting “extreme problem” or “some problem” in all five dimensions of the EQ-5D index compared with those without comorbidity (OR: mobility, 3.99, 6.22; usual activity, 2.67, 5.43; self-care, 2.60, 3.95; pain or discomfort, 2.22, 3.44; anxiety or depression, 1.75, 2.45). The number of comorbidities was found to be negatively correlated with the HRQoL. Conclusion: Prevalent comorbidities were found to be the significant underlying cause of declined HRQoL. To raise diabetes awareness and for better disease management, the exposition of comorbidities in regards to HRQoL of diabetic people should be considered for type 2 diabetes management schemas.”

Regarding Introduction. The introduction is fair. Please be mindful on the punctuation in the sentences. For example, there is a comma after “To our knowledge, this…. ” I suggest the term “To the best of our knowledge,”:

“To the best of our knowledge, this study pioneers in evaluating the relationship between HRQoL and 107 comorbidities using the EQ-5D-5L instrument in the type 2 diabetes population in the southern 108 regions of Bangladesh.” It should be ‘regions’ and ‘not region’.”

It was mentioned in line 76 and 77 that, “Apart from the economical standpoints, the mere existence of diabetes also takes a psychological toll on the individual diagnosed, due to this disease’s chronic manifestations and auxiliary clinical complications.” Please change “economical” to ‘economic”. Please consider the following references for this statement, which is currently without any citations.

Mohammad Farris Iman Leong Abdullah, Hatta Sidi, Arun Ravindran, Paula Junggar Gosse, Emily Samantha Kaunismaa, Roslyn Laurie Mainland, Norlaila Mustafa, Nurul Hazwani Hatta, Puteri Arnawati, Amelia Yasmin Zulkifli, Luke Sy-Cherng Woon. How Much Do We Know About the Biopsychosocial Predictors Glycaemic Control? Age and Clinical Factors Predict Glycaemic Control, But Psychological Factors Do Not. J Diabetes Res. 2020; 2020: 2654208. Online 2020 May 1. doi: 10.1155/2020/2654208

Luke Sy-Cherng Woon, Hatta Sidi, Arun Ravindran, Paula Junggar Gosse, Roslyn Laurie Mainland, Emily Samantha Kaunismaa, Nurul Hazwani Hatta, Puteri Arnawati, Amelia Yasmin Zulkifli, Norlaila Mustafa, Mohammad Farris Iman Leong Abdullah. Depression, anxiety, and associated factors in patients with diabetes: evidence from the anxiety, depression, and personality traits in diabetes mellitus (ADAPT-DM) study. BMC Psychiatry. 2020; 20: 227. Online 2020 May 12. doi: 10.1186/s12888-020-02615-y

Luke Sy-Cherng Woon, Roslyn Laurie Mainland, Emily Samantha Kaunismaa, Paula Junggar Gosse, Arun Ravindran & Hatta Sidi. What makes poor diabetic control worse? A cross-sectional survey of biopsychosocial factors among patients with poorly controlled diabetes mellitus in Malaysia. Int J Diabetes Mellitus in Dev Countries. 2021. DOI: 10.1007/s13410-020-00918-0

Regarding methods: The methods are okay, except for the missing sampling strategy. Is there any cluster and stratified sampling performed for the two southern hospitals in Bangladesh?

Please do a correlation (Pearson/Spearman) study between all medical comorbidities and the measuring scales for quality of life.

Regarding results. The Tables are self-explanatory and relevant.

Regarding discussion. Please avoid the language of causality in your discussion as the study performed was cross-sectional.

The first paragraph stated, “Patients with diabetes mellitus go through various degrees of HRQoL because of the chronic 308 nature of the disease and different levels of compliance followed over time. HRQoL deteriorates 309 significantly if they suffer from multiple comorbidities along with diabetes compared to the 310 patients who didn’t have any comorbidity.” Based on your literature findings and clinical experience, what would the factors associated with poorer HRQoL and multiple medical comorbidities besides the confounding effect of all oxidative diseases? Could it be a personality factor? Or any comorbidities with severe mental health problems?

Luke S Woon, Hatta Sidi, Norlaila M. Factor structure of the Malay-version Generalized Anxiety Disorder-7 (GAD-7) Questionnaire among Patients with Daibetes Mellitus. Med and Health. June 2020: 15(1): 208 - 217.

Reviewer #2: Study on effects of comorbidities on health-related quality of life among type 2 diabetes mellitus patients is an important area of research in diabetes care. However, there are few queries that need to be addressed by the authors.

1. Were all the adults type 2 diabetes mellitus patients attending the

in patients department and out patients department of the selected public and private hospitals recruited for the study?

2. How was the sample size of 2136 diabetes patients determined?

3. How were the types of comorbidities included in the study arrived at?

4. Though as mentioned in the manuscript that “the principal aim is to investigate impact of comorbidities based on their number and nature”, there is no finding on the impact of different comorbidity or nature of comorbidity.

5. Similarly, there is no finding on the different multimorbidity pattern in the manuscript, though it is one of the objectives.

6. Model 1 and Model 2 need to be mentioned.

7. Questionnaire used in the study for demography and comorbidity assessment maybe uploaded.

6. PLOS authors have the option to publish the peer review history of their article (what does this mean?). If published, this will include your full peer review and any attached files.

Reviewer #1: **Yes: **Hatta Sidi

Reviewer #2: No

---

## [Author Response · Author response to Decision Letter 0]

21 Feb 2022

Additional Editor Comments:

1. Please ensure that your manuscript meets PLOS ONE's style requirements, including those for file naming and tables. The PLOS ONE style templates can be found at:

Response

The guideline has been followed accordingly. 

2. Please ensure that your tables are formatted according to PLOS ONE's style requirements which can be found at:

https://journals.plos.org/plosone/s/tables

Response

Tables have been formatted as per instruction.

3. Please seek assistance from a native English speaker or professional English editing service to proofread your revised manuscript as there are some grammatical errors which need to be corrected.

Response

A native English speaker has checked and modified the manuscript. Grammatical errors have been fixed. (line 59, 109-111, 138, 160-162, 183, 204, 255-260, 338, 388).

Reviewer #1: Effect of comorbidities on health-related quality of life among adults with type 2 diabetes attending different hospitals in southern Bangladesh

Title: The title is misleading because there is an implication of the language of causality. Please rephrase the title to: “The relationship between medical comorbidities and health-related quality of life among adults with type 2 diabetes: The experience of different hospitals in southern Bangladesh”.

Response:

Authors would like to thank the learned reviewer for this valuable feedback. Title has been changed as suggessted.

Overall comments:

The writing of the manuscript is good, with fair scientific merits based on the statistical methods employed. 

• Having said that, there were some concerns of inference on the statistics of “correlation” when the author(s) mentioned that “The number of comorbidities was found to be negatively correlated with the HRQoL.” Has any correlation study been employed in this research? The authors may change the wording of the sentences or employ the relevant statistics and discuss the findings.

Response: 

With the above sentence, we actually referred to the statistically significant relationships between the number of comorbidities and quality of life that was found from the three estimated models shown in Table 3. The sentence has been rephrased as advised (lines: 49-50).

• Regarding the use of the statistical inference, please spell out OR as an odds ratio. The 95% CI was missing at each OR. Please insert it and standardise it through the text.

 Response: 

 Revisions have been made as advised, in the abstract (lines: 46-49) and in the results sections (lines: 246-256). 

Specific comments:

Regarding abstract:

The abstract needs some revisions, as it contains the language of causality in the objective. The writing style was also required to be refined. I suggest some changes:

"Objective Health-related quality of life (HRQoL) is a critical determinant to assess the severity of chronic diseases like diabetes mellitus. It has a close association with complications, comorbidities, and medical aid. This study aimed to estimate the prevalence of medical comorbidities and determine the relationship between comorbidities and HRQoL among type 2 diabetic patients of southern Bangladesh. Method: This study was a cross-sectional study conducted through face-to-face interviews using a pre-tested structured questionnaire and by reviewing patient’s health records with prior written consent. The study was conducted on 2,136 patients with type 2 diabetes attending five hospitals of Chattogram, Bangladesh, during the tenure of November 2018 to July 2019. Quality of life was measured using the widely-used index of EQ-5D that considers 243 different health states and uses a scale in which 0 indicates a health state equivalent to death and 1 indicates perfect health status. The five dimensions of the quality index included mobility, self-care, usual activities, pain or discomfort, and anxiety or depression. Results: Patients with three comorbidities and with four or more comorbidities had a higher probability of reporting “extreme problem” or “some problem” in all five dimensions of the EQ-5D index compared with those without comorbidity (OR: mobility, 3.99, 6.22; usual activity, 2.67, 5.43; self-care, 2.60, 3.95; pain or discomfort, 2.22, 3.44; anxiety or depression, 1.75, 2.45). The number of comorbidities was found to be negatively correlated with the HRQoL. Conclusion: Prevalent comorbidities were found to be the significant underlying cause of declined HRQoL. To raise diabetes awareness and for better disease management, the exposition of comorbidities in regards to HRQoL of diabetic people should be considered for type 2 diabetes management schemas.”

Response: 

Changes have been made accordingly (line 28-55). 

Regarding Introduction. The introduction is fair. Please be mindful on the punctuation in the sentences. For example, there is a comma after “To our knowledge, this…. ” I suggest the term “To the best of our knowledge,”:

“To the best of our knowledge, this study pioneers in evaluating the relationship between HRQoL and 107 comorbidities using the EQ-5D-5L instrument in the type 2 diabetes population in the southern 108 regions of Bangladesh.” It should be ‘regions’ and ‘not region’.”

Response: 

Changes have been made accordingly (Line 106).

It was mentioned in line 76 and 77 that, “Apart from the economical standpoints, the mere existence of diabetes also takes a psychological toll on the individual diagnosed, due to this disease’s chronic manifestations and auxiliary clinical complications.” Please change “economical” to ‘economic”. Please consider the following references for this statement, which is currently without any citations.

Response: 

Changes have been made accordingly.

Mohammad Farris Iman Leong Abdullah, HattaSidi, ArunRavindran, Paula Junggar Gosse, Emily Samantha Kaunismaa, Roslyn Laurie Mainland, Norlaila Mustafa, NurulHazwaniHatta, PuteriArnawati, Amelia YasminZulkifli, Luke Sy-CherngWoon. How Much Do We Know About the Biopsychosocial Predictors Glycaemic Control? Age and Clinical Factors Predict Glycaemic Control, But Psychological Factors Do Not. J Diabetes Res. 2020; 2020: 2654208. Online 2020 May 1. doi: 10.1155/2020/2654208

Luke Sy-CherngWoon, HattaSidi, ArunRavindran, Paula Junggar Gosse, Roslyn Laurie Mainland, Emily Samantha Kaunismaa, NurulHazwaniHatta, PuteriArnawati, Amelia YasminZulkifli, Norlaila Mustafa, Mohammad Farris Iman Leong Abdullah. Depression, anxiety, and associated factors in patients with diabetes: evidence from the anxiety, depression, and personality traits in diabetes mellitus (ADAPT-DM) study. BMC Psychiatry.2020; 20: 227. Online 2020 May 12. doi: 10.1186/s12888-020-02615-y

Luke Sy-CherngWoon, Roslyn Laurie Mainland, Emily Samantha Kaunismaa, Paula Junggar Gosse, ArunRavindran&HattaSidi. What makes poor diabetic control worse? A cross-sectional survey of biopsychosocial factors among patients with poorly controlled diabetes mellitus in Malaysia. Int J Diabetes Mellitus in Dev Countries. 2021. DOI: 10.1007/s13410-020-00918-0

Response: 

Authors would like to thank for these suggestions. Suggested references have been added (Reference 8-10).

Regarding methods: The methods are okay, except for the missing sampling strategy. Is there any cluster and stratified sampling performed for the two southern hospitals in Bangladesh?

Response: The data were collected adopting a convenience sampling strategy. Revisions made in the method section (lines: 123-127).

Please do a correlation (Pearson/Spearman) study between all medical comorbidities and the measuring scales for quality of life.

Response: 

Pearson’s chi-square tests performed and revisions made in the results section (line: 268-271).

Regarding results. The Tables are self-explanatory and relevant.

Response: 

Authors would like to thank the distinguished reviewer for positive feedback. 

Regarding discussion. Please avoid the language of causality in your discussion as the study performed was cross-sectional.

Response: We have carefully checked the article for not using the language of causality, rather we have mentioned this as a limitation that the study is no way meant to establish causal relationship given its cross-sectional design. 

The first paragraph stated, “Patients with diabetes mellitus go through various degrees of HRQoL because of the chronic 308 nature of the disease and different levels of compliance followed over time. HRQoL deteriorates 309 significantly if they suffer from multiple comorbidities along with diabetes compared to the 310 patients who didn’t have any comorbidity.” Based on your literature findings and clinical experience, what would the factors associated with poorer HRQoL and multiple medical comorbidities besides the confounding effect of all oxidative diseases? Could it be a personality factor? Or any comorbidities with severe mental health problems?

Response: 

Thank you for raising this excellent point. Mental health problems are integral part of HRQoL and personality factors are also intrinsic indeed. For many societies including in Bangladesh certain mental health illnesses like anxiety and depression are not perceived as mainstream health problems. We have, however, broadly covered neurological disorders as part of comorbidity assessment, which was reported by a good number of participants. 

Luke S Woon, HattaSidi, Norlaila M. Factor structure of the Malay-version Generalized Anxiety Disorder-7 (GAD-7) Questionnaire among Patients with Daibetes Mellitus. Med and Health. June 2020: 15(1): 208 – 217.

Response: 

The reference has been added in the discussion section (Reference 10). 

Reviewer #2: 

Study on effects of comorbidities on health-related quality of life among type 2 diabetes mellitus patients is an important area of research in diabetes care. However, there are few queries that need to be addressed by the authors.

1. Were all the adults type 2 diabetes mellitus patients attending the in patients department and out patients department of the selected public and private hospitals recruited for the study?

Response: 

Adult type 2 diabetes mellitus patients who satisfy the selection criteria were recruited using consecutive sampling methods from the selected hospital or clinic’s outpatients departments. Line 123-125 

2. How was the sample size of 2136 diabetes patients determined?

Response: 

Factors considered for sample size calculation have been narrated in the sub-section 2.1 under the Methodology section, lines 125-129. 

3. How were the types of comorbidities included in the study arrived at?

Response: 

Types of comorbidities were included based on literatures that we have reviewed in this regard. References of those literatures are cited as number 16, 17, 18, and 19 in the introduction section. The prevalence of the comorbidities was determined from data reported by the participants and varied from their diabetes handbook, doctors prescriptions. 

4. Though as mentioned in the manuscript that “the principal aim is to investigate impact of comorbidities based on their number and nature”, there is no finding on the impact of different comorbidity or nature of comorbidity.

Response: 

We have edited the primary objective as stated in earlier version, now more aligned with what we have measured and analyzed to estimate the prevalence and determinants of comorbidities on HRQoL as mentioned in the abstract and lines 109-111. 

5. Similarly, there is no finding on the different multimorbidity pattern in the manuscript, though it is one of the objectives.

Response: It has been mentioned in the modified manuscript (line 255-260).

6. Model 1 and Model 2 need to be mentioned.

Response: 

Model 1 and Model 2 has been mentioned as advised in the results section (lines: 276-281). 

7. Questionnaire used in the study for demography and comorbidity assessment maybe uploaded.

Response: 

Questionnaire has been uploaded.

---

## [Decision Letter · Decision Letter 1]

16 Mar 2022

PONE-D-21-37788R1The relationship between medical comorbidities and health-related quality of life among adults with type 2 diabetes: The experience of different hospitals in southern BangladeshPLOS ONE

Dear Dr. Mannan,

Thank you for submitting your manuscript to PLOS ONE. After careful consideration, we feel that it has merit but does not fully meet PLOS ONE’s publication criteria as it currently stands. Therefore, we invite you to submit a revised version of the manuscript that addresses the points raised during the review process.

We look forward to receiving your revised manuscript.

Kind regards,

Mohammad Farris Iman Leong Bin Abdullah, Dr. Psych

Academic Editor

PLOS ONE

Journal Requirements:

Reviewers' comments:

Reviewer's Responses to Questions

**Comments to the Author**

1. If the authors have adequately addressed your comments raised in a previous round of review and you feel that this manuscript is now acceptable for publication, you may indicate that here to bypass the “Comments to the Author” section, enter your conflict of interest statement in the “Confidential to Editor” section, and submit your "Accept" recommendation.

Reviewer #1: All comments have been addressed

Reviewer #2: All comments have been addressed

2. Is the manuscript technically sound, and do the data support the conclusions?

Reviewer #1: Yes

Reviewer #2: Yes

3. Has the statistical analysis been performed appropriately and rigorously? 

Reviewer #1: Yes

Reviewer #2: I Don't Know

4. Have the authors made all data underlying the findings in their manuscript fully available?

Reviewer #1: Yes

Reviewer #2: Yes

5. Is the manuscript presented in an intelligible fashion and written in standard English?

Reviewer #1: Yes

Reviewer #2: Yes

6. Review Comments to the Author

Reviewer #1: There was an improvement in the writing and the presentation of the revised manuscript. However, some comments need to be addressed, and further edits are necessary.

For the discussion, it was stated that: "Patients with diabetes mellitus go through various degrees of HRQoL because of the chronic nature of the disease and different levels of compliance followed over time." These sentences are incomplete. Is it low or high HRQoL? For " HRQoL deteriorates significantly if they suffer from multiple comorbidities along with diabetes compared to the 331 patients who didn’t have any comorbidity." , I think the study does not have enough evidence to lead to a conclusion on the cause and effect, i.e., chronic disease lead to a low HRQoL, and vice-versa. The bilateral correlation must be discussed in depth. For example, people with low HRQoL may neglect their health and medication, leading o further deterioration of their blood sugar levels and other chronic medical problems (hyperlipidemia, obesity, etc.).

The discussion on the recommendation for intervention was also inadequate and needed elaboration. For example, how those affected individuals may improve their quality of life besides taking medication? What food is necessary to improve blood sugar and cholesterol level. How about a high fiber diet and some vegetables like ladyfinger and a low glycemic index food.

Reviewer #2: All the comments raised were satisfactorily addressed by the authors. No further query raised by this reviewer.

7. PLOS authors have the option to publish the peer review history of their article (what does this mean?). If published, this will include your full peer review and any attached files.

Reviewer #1: No

Reviewer #2: **Yes: **Sandipana Pati

---

## [Author Response · Author response to Decision Letter 1]

23 Mar 2022

Reviewer #1: 

There was an improvement in the writing and the presentation of the revised manuscript. However, some comments need to be addressed, and further edits are necessary.

For the discussion, it was stated that: "Patients with diabetes mellitus go through various degrees of HRQoL because of the chronic nature of the disease and different levels of compliance followed over time." These sentences are incomplete. Is it low or high HRQoL? 

Response: Thank you for the comment. We have restated the sentences between lines 312-317.

For " HRQoL deteriorates significantly if they suffer from multiple comorbidities along with diabetes compared to the 331 patients who didn’t have any comorbidity.", I think the study does not have enough evidence to lead to a conclusion on the cause and effect, i.e., chronic disease lead to a low HRQoL, and vice-versa. 

Response: We have revisited the interpretation following suggestion of the learned reviewer in lines 314-316. 

The bilateral correlation must be discussed in depth. For example, people with low HRQoL may neglect their health and medication, leading o further deterioration of their blood sugar levels and other chronic medical problems (hyperlipidemia, obesity, etc.).

Response: We have expanded discussion on bilateral and multilateral correlation of HRQoL and Comorbidities in Lines 317-322.

The discussion on the recommendation for intervention was also inadequate and needed elaboration. For example, how those affected individuals may improve their quality of life besides taking medication? What food is necessary to improve blood sugar and cholesterol level. How about a high fiber diet and some vegetables like ladyfinger and a low glycemic index food.

Response: We have added text highlighting the importance of healthy diet and food with low glycemic index as recommendation in lines 386-388.

Reviewer #2: 

All the comments raised were satisfactorily addressed by the authors. No further query raised by this reviewer.

Response: We would like to thank the honorable reviewer for positive feedback and response. .

---

## [Decision Letter · Decision Letter 2]

14 Apr 2022

The relationship between medical comorbidities and health-related quality of life among adults with type 2 diabetes: The experience of different hospitals in southern Bangladesh

PONE-D-21-37788R2

Dear Dr. Mannan,

We’re pleased to inform you that your manuscript has been judged scientifically suitable for publication and will be formally accepted for publication once it meets all outstanding technical requirements.

Kind regards,

Mohammad Farris Iman Leong Bin Abdullah, Dr Psych

Academic Editor

PLOS ONE

Additional Editor Comments (optional):

Reviewers' comments:

Reviewer's Responses to Questions

**Comments to the Author**

1. If the authors have adequately addressed your comments raised in a previous round of review and you feel that this manuscript is now acceptable for publication, you may indicate that here to bypass the “Comments to the Author” section, enter your conflict of interest statement in the “Confidential to Editor” section, and submit your "Accept" recommendation.

Reviewer #1: All comments have been addressed

2. Is the manuscript technically sound, and do the data support the conclusions?

Reviewer #1: Yes

3. Has the statistical analysis been performed appropriately and rigorously? 

Reviewer #1: Yes

4. Have the authors made all data underlying the findings in their manuscript fully available?

Reviewer #1: Yes

5. Is the manuscript presented in an intelligible fashion and written in standard English?

Reviewer #1: Yes

6. Review Comments to the Author

Reviewer #1: All comments have been addressed by the authors. Improvements have been performed in the discussion part.

7. PLOS authors have the option to publish the peer review history of their article (what does this mean?). If published, this will include your full peer review and any attached files.

Reviewer #1: No

---

## [Editor Report · Acceptance letter]

16 May 2022

PONE-D-21-37788R2 

The relationship between medical comorbidities and health-related quality of life among adults with type 2 diabetes: The experience of different hospitals in southern Bangladesh 

Dear Dr. Mannan:

I'm pleased to inform you that your manuscript has been deemed suitable for publication in PLOS ONE. Congratulations! Your manuscript is now with our production department. 

Kind regards, 

on behalf of

Dr. Mohammad Farris Iman Leong Bin Abdullah 

Academic Editor

PLOS ONE